# Global Proteomics for Identifying the Alteration Pathway of Niemann–Pick Disease Type C Using Hepatic Cell Models

**DOI:** 10.3390/ijms242115642

**Published:** 2023-10-27

**Authors:** Keitaro Miyoshi, Eiji Hishinuma, Naomi Matsukawa, Yoshitaka Shirasago, Masahiro Watanabe, Toshihiro Sato, Yu Sato, Masaki Kumondai, Masafumi Kikuchi, Seizo Koshiba, Masayoshi Fukasawa, Masamitsu Maekawa, Nariyasu Mano

**Affiliations:** 1Faculty of Pharmaceutical Sciences, Tohoku University, 1-1 Seiryo-machi, Aoba-Ku, Sendai 980-8574, Japan; 2Advanced Research Center for Innovations in Next-Generation Medicine, Tohoku University, 2-1 Seiryo-machi, Aoba-Ku, Sendai 980-8573, Japan; ehishi@ingem.oas.tohoku.ac.jp (E.H.);; 3Tohoku Medical Megabank Organization, Tohoku University, 2-1 Seiryo-machi, Aoba-Ku, Sendai 980-8573, Japan; 4Department of Biochemistry and Cell Biology, National Institute of Infectious Diseases, 1-23-1, Toyama, Shinjuku-ku, Tokyo 162-8640, Japan; 5Graduate School of Pharmaceutical Sciences, Tohoku University, 1-1 Seiryo-machi, Aoba-Ku, Sendai 980-8574, Japan; 6Department of Pharmaceutical Sciences, Tohoku University Hospital, 1-1 Seiryo-machi, Aoba-Ku, Sendai 980-8574, Japan

**Keywords:** Niemann–Pick disease type C, global proteomics, liquid chromatography–electrospray ionization tandem mass spectrometry, model cell, knock out, enrichment pathway analysis

## Abstract

Niemann–Pick disease type C (NPC) is an autosomal recessive disorder with progressive neurodegeneration. Although the causative genes were previously identified, NPC has unclear pathophysiological aspects, and patients with NPC present various symptoms and onset ages. However, various novel biomarkers and metabolic alterations have been investigated; at present, few comprehensive proteomic alterations have been reported in relation to NPC. In this study, we aimed to elucidate proteomic alterations in NPC and perform a global proteomics analysis for NPC model cells. First, we developed two NPC cell models by knocking out *NPC1* using CRISPR/Cas9 (KO1 and KO2). Second, we performed a label-free (LF) global proteomics analysis. Using the LF approach, more than 300 proteins, defined as differentially expressed proteins (DEPs), changed in the KO1 and/or KO2 cells, while the two models shared 35 DEPs. As a bioinformatics analysis, the construction of a protein–protein interaction (PPI) network and an enrichment analysis showed that common characteristic pathways such as ferroptosis and mitophagy were identified in the two model cells. There are few reports of the involvement of NPC in ferroptosis, and this study presents ferroptosis as an altered pathway in NPC. On the other hand, many other pathways and DEPs were previously suggested to be associated with NPC, supporting the link between the proteome analyzed here and NPC. Therapeutic research based on these results is expected in the future.

## 1. Introduction

Niemann–Pick disease type C (NPC) is a progressive and life-limiting autosomal recessive disorder characterized by progressive neurodegeneration [1,2,3]. Due to the development of laboratory medicines for NPC [2,4,5,6,7,8,9], the number of NPC patients is increasing, and the current estimated prevalence is approximately 1/100,000 [10]. NPC is caused by mutations in *NPC1* or *NPC2*. *NPC1* codes for the NPC1 cholesterol transport membrane protein in lysosomes, whereas *NPC2* codes for the NPC2 intracellular cholesterol-binding soluble protein in lysosomes [11]. NPC1 and NPC2 coordinate in the transport of cholesterol, and *NPC1* or *NPC2* mutations can cause cholesterol traffic dysfunction [12,13]. Although NPC is not related to enzyme deficiency, it is considered a lysosomal storage disease (LSD). In addition, various sphingolipids, including sphingomyelin [14,15] and glycosphingolipids, [16,17] accumulate in NPC. Glycosphingolipid metabolism is a therapeutic target pathway in NPC, and miglustat, which inhibits glucosylceramide synthase, is currently the only approved drug [18].

The pathophysiology of NPC involves patients presenting a variety of symptoms. The onset age of NPC symptoms varies from the neonatal to adulthood, and the clinical symptoms also differ widely [1,2,10]. Hepatosplenomegaly and cholestasis are typical symptoms, and many patients develop various neurodegenerative disorders depending on their age [1]. Disease prognosis is generally improved when treatment begins earlier [18]; therefore, earlier diagnosis is desired, and a number of clinical biomarkers have been developed for the early discovery of this condition [4,5,6,7]. Since 2010, oxysterols [19,20], lysosphingomyelin [14,21], *N*-palmitoyl-*O*-phosphocholine-serine [22,23,24,25] (previously called Lyso-SM-509), and abnormal bile acids containing conjugates [8,9,26,27,28,29,30,31,32,33] have been reported as biomarkers of NPC. However, the detailed mechanisms underlying the progression of this disease remain unknown [10].

Proteomics is an omics analytical technique used to elucidate the molecular and pathophysiological alterations in expressed cellular proteins [34,35,36,37,38,39,40,41,42]. Although the genome is static, the proteome is dynamic [43,44,45] and has a longer turnover time than the metabolome [46,47]. Liquid chromatography–tandem mass spectrometry (LC/MS/MS) is commonly used for proteomic analyses [41,48,49,50], particularly in the form of nanoLC/MS/MS [41,50,51]. In searching for the alteration pathways of various diseases, an untargeted proteomics approach called global proteomics has been widely used [41,52].

In this study, to reveal the pathophysiology in the proteomic aspects of NPC and discover novel therapeutic targets, we aimed to elucidate the altered pathway in NPC model cells using the global proteomics approach.

## 2. Results and Discussion

### 2.1. NPC Model Cell Development

Gene editing was performed on *NPC1* by knocking out *NPC1* using the CRISPR/Cas9 method [53,54,55]. We attempted to target two different gene sites simultaneously. One approach focused on sites A and C (for KO1 cells), and the other focused on sites B and D (for KO2 cells). Sites A and B were located in the signal sequences, Site C was located on exon 8, which is included in the first extracellular loop, and Site D was located on exon 22, which is in ninth transmembrane region. As a result of the mutation, NPC1 proteins were not detected via immunoblotting in either KO1 or KO2 (Appendix A). Filipin staining, which is the gold-standard method for NPC diagnosis of patients [56], was then performed, and cholesterol accumulation was observed in both the KO1 and KO2 cells (Appendix A). In addition, enzymatic free-cholesterol quantitation showed a significant difference between the KO and WT cells (Figure 1).

NPC is caused by a lack of function of NPC1 or NPC2 [1]. A representative NPC cell phenotype is the accumulation of lysosomal free cholesterol [12,13,57]. In the present study, cell DNA editing engineering was successful (Appendix A), and two types of NPC1 model cells were developed (KO1 and KO2). In the KO1 cells, mutations of exon 1 and exon 8 were targeted. Exon 8 contains the second transmembrane region and the 1first extracellular loop (Appendix A). Therefore, gene editing provided an NPC 1 protein that was undetectable via immunoblotting with the NPC1 antibody. In the KO2 cells, mutations in the signal sequences on exon 1 and the ninth transmembrane region on exon 22 were identified (Appendix A). Similar to the KO1 cells, we succeeded in developing NPC1 mutant cells. Both filipin staining and enzymatic quantification provided the representative NPC phenotypes.

### 2.2. Label-Free Global Proteomics

We then performed a global proteomics analysis known as the label-free (LF) method. The global method is a nontargeted approach that does not define specific molecules as analytes and covers a wide range of molecule sizes.

First, we summarized the protein numbers identified from the LF global proteomics analysis (Figure 2, Appendix A). The regions that were not overlaid with any other regions contained a few proteins. Each cell line was analyzed in five biological replicates, and proteins identified in four or more replicates were used for a subsequent pathway analysis. As a result, a total of 3331 proteins were identified in all cell types, 3390 proteins were identified in the WT and KO1 cells, 3434 proteins were identified in the WT and KO2 cells (Figure 2).

Second, we investigated the characteristics of each sample using a multivariate analysis. A principal component analysis (PCA) was first performed using 3331 proteins whereby three samples (WT, KO1, and KO2) were plotted closely for each group. In addition, the KO1 and KO2 cells were shown in different areas (Figure 3a). This indicates that the characteristics of the NPC1 model cells are different from those of WT cells, and the proteins expressed in the KO1 and KO2 cells also differ from each other. NPC1 showed clinically diverse phenotypes [1,2,10]. In NPC patients, the gene mutations of *NPC1* vary by race [58,59,60,61] and various gene mutations are known, with novel mutations currently being reported [62]. However, a detailed correlation between gene mutations and clinical phenotypes remains unknown [10]. In this study, the KO1 cells, which have two mutation sites on the signal sequence on exon 1 and the first extracellular loop on exon 8, and the KO2 cells, which have two mutation sites on the signal sequence on exon 1 and the ninth transmembrane region on exon 22, showed different results in the PCA. We also utilized a loading plot in order to infer which proteins are responsible for the difference in each type of cell (Figure 3b). In the loading plot, the cumulative contribution ratio of the first and second principal components (PC1 and PC2) was 93.3%, which is a large amount of information. Keratin, type II cytoskeletal 8 (KRT8), Keratin, type I cytoskeletal 18 (KRT18), 60 kDa heat shock protein, mitochondrial (HSPD1), Actin, cytoplasmic 1 (ACTB), and Vimentin (VIM) represent some of the largest absolute values of the major components of both PC1 and PC2. These proteins could be responsible for the differences among cell types.

Finally, we created a volcano plot which shows the degree of change and statistical significance simultaneously on the graph [63,64,65] (Figure 4). The selection criteria for differentially expressed proteins (DEPs) were set at a two-fold change and statistical significance (*p* < 0.05) between the NPC1 model and WT cells. The identified DEPs are summarized in Appendix A, Table 1, and Appendix A. In Appendix A, the total numbers of DEPs which were both up- and downregulated in the NPC1 model cells are shown. The overlap of the two model cells is presented in Table 1. Twenty-five proteins were upregulated in both the KO1 and KO2 cells, and 10 proteins were downregulated. Two proteins increased in KO1 cells but decreased in KO2 cells, and four proteins decreased in both KO1 and KO2 cells. Other proteins were either upregulated or downregulated in KO1 or KO2 cells only. The overlapping protein numbers were low among the total DEPs. The common DEPs in the KO1 and KO2 cells were not the main DEPs observed. In addition, there were proteins that were upregulated in the KO1 cells but were downregulated in the KO2 cells and vice versa. As mentioned above, the NPC phenotype is diverse, and patients with NPC have varied symptoms [1,2]. Therefore, the proteomic results for the KO1 and KO2 cells indicate that the difference in mutation sites between the KO1 and KO2 cells affects the proteomic expression pattern.

### 2.3. Bioinfomatics Analysis

We evaluated the interaction of the DEPs using the Search Tool for the Retrieval of Interacting Genes/Proteins (STRING). The DEPs of the KO1 cells provided us with a protein–protein interaction (PPI) network consisting of 136 nodes and 154 edges. The expected number of edges from the number of nodes was 96, which is a much lower value than the actual edges, and the PPI enrichment *p* value was 3.36 × 10^−8^ (Appendix A). The DEPs of the KO2 cells provided us with a PPI network consisting of 249 nodes and 855 edges. The expected number of edges from the number of nodes was 452, which is a much lower value than the actual edges, and the PPI enrichment *p* value was <1.0 × 10^−16^ (Appendix A). The nodes of the upregulated proteins in the DEPs were displayed in red, and downregulated proteins were displayed in green. These results show the overall picture of up/downregulation and protein–protein interactions in the DEPs and suggest that the DEPs have strong interactions in the KO1 and KO2 cells. In addition, an analysis excluding text mining from the interaction sources showed 71 edges in the KO1 cells and 518 edges in the KO2 cells, with percentages of 46.1% and 60.6%, respectively. It was confirmed that there were enough reliable edges other than text mining, such as curated databases, experiments, and co-expression.

In addition, we displayed the proteins coded via ferroptosis in blue and those coded via ribosome biogenesis in eukaryotes in yellow. These pathways were identified as particularly significant in the enrichment analysis (ferroptosis was the highest in both KO1 and KO2 cells, and ribosome biogenesis in eukaryotes was the second highest in the KO2 cells, Figure 5). In particular, ferroptosis was identified for both KO1 and KO2 cells. Marked proteins are linked by many edges and are in regions with many edges in the figure, suggesting strong interactions of the DEPs in each cell at these pathways (Appendix A).

We performed an enrichment analysis, which is used for finding significant functions and pathways [38,40,41,66]. We used data from the Gene Ontology database (GO) and the Kyoto Encyclopedia of Genes and Genomes pathway database (KEGG) [67,68]. In the GO enrichment analysis, the results suggested that the DEPs in the KO1 cells are involved in biological processes (BPs) such as autophagy, cellular components (CCs) such as autolysosomes, and molecular function (MFs) such as RNA binding. Similarly, relationships with many functions were suggested for the KO2 cells (Appendix A, Appendix A). In the KEGG pathway enrichment analysis, seven and ten pathways were selected from the KO1 and KO2 cells as significant, respectively (Figure 5). From the KEGG pathway analysis results for both the KO1 and KO2 cells, ferroptosis (ranked No. ranked in both KO1 and KO2 cells), lysosome (ranked No. 4 in KO1 cells and ranked No.3 in KO2 cells), mitophagy (ranked No. 2 in KO1 cells and ranked No. 5 in KO2 cells), and metabolic pathways (ranked No. 7 in KO1 cells and ranked No. 8 in KO2 cells) were identified as significantly changed.

The common DEPs in the KO1 and KO2 cells were not the main DEPs observed. As mentioned above, the NPC phenotype is diverse, and patients with NPC have varied symptoms [1,2]. Conversely, the DEPs identified in both KO1 and KO2 cells could be relatively ubiquitous DEPs in NPC patients, making them more important for understanding the pathogenesis of NPC and searching for therapeutic targets. The proteins identified as DEPs and pathways in both KO1 and KO2 cells are mainly described below.

Ferroptosis is a cell death pathway catalyzed by Fe^2+^ ions and lipid peroxidation [69]. Ferroptosis has been reported to be involved in various neurodegenerative diseases, including amyotrophic lateral sclerosis, Parkinson’s disease, and Alzheimer’s disease [70]. However, the relationship between ferroptosis and NPC remains unclear. Regarding Fe^2+^ metabolism, abnormalities in iron homeostasis in NPC mouse models have been reported [71]. In another report, the degradation of ferritin via intracellular autophagy and the progression of ferroptosis were also identified [72]. In an enrichment analysis using the KEGG, ferroptosis was identified with increased nuclear receptor coactivator 4 (NCOA4), ferritin light chain (FTL), ferritin heavy chain (FTH1), and microtubule-associated proteins 1A/1B light chain 3B (MAP1LC3B). Of the four proteins identified in both KO1 and KO2 cells, NCOA4 is the ferritin cargo receptor required for transport into the lysosome [73,74], targeting ferritin in the autophagosome via a selective autophagic process, ferritinophagy. Both light (FTL) and heavy-chain (FTH1) subunits comprise 24-subunit spherical shell protein ferritin [75]. MAP1LC3B is known as a representative human Atg8 orthologue which binds to cargo receptors on the surfaces of autophagosomes [73]. NCOA4, FTL, FTH1, and MAP1LC3B, identified in both KO1 and KO2 cells, are all specifically related to ferritinophagy [75,76], the autophagy of ferritin for iron homeostasis [76,77]. The degradation of ferritin via ferritinophagy induces Fenton reactions by releasing ferric iron ions, resulting in the progression of ferroptosis [74,78], but it remains unclear whether increases in these DEPs cause ferritinophagy and ferroptosis or not.

For BP, the term of cellular iron ion homeostasis was identified as the seventh at KO1 and the synonymous term of iron ion homeostasis was identified as the fifth at KO2, and among the DEPs, NCOA4, FTL, and FTH1 increased and ATP-binding cassette sub-family B member 6 (ABCB6) decreased. The term represents all the processes involved in maintaining the internal steady state of iron ions at the cellular level. ABCB6 is an energy-dependent porphyrin transporter that catalyzes porphyrin transport from the cytosol to the extracellular fluid and from the cytosol to mitochondria, thereby contributing to heme biosynthesis and iron homeostasis. It is involved in the regulation of heme biosynthesis and iron homeostasis [79]. It is known that ABCB6 promote cellular defense responses against various toxic insults, so significant decreases in it in the KO1 and KO2 cells could promote the production of ROS [80,81]. In addition, the term of the intracellular sequestering of iron ion was identified in the KO1 cells (also ranked and identified in the KO2 cells with *p* = 0.072), the process through which iron ions are bound or sequestered intracellularly and separated from the other components of the biological system. From the above, many DEPs and pathways related to ferroptosis were extracted, indicating a link between ferroptosis and NPC which has been rarely reported.

NPC is a type of LSD, and changes in the expression of various proteins in lysosomes have been reported as the respective substrates accumulate in lysosomes due to deficiencies in lysosome hydrolases or transporters as well as NPC1 [10,11,12,13]. Lysosomes were identified via the KEGG analysis in both KO1 and KO2 cells, with decreased lysosomal acid glucosylceramidase (GBA) and lysosomal alpha-mannosidase (MAN2B1) and increased CD63 antigen (CD63). For CCs, autolysosomes were identified at the second and thirteenth positions in the KO1 and KO2 cells, respectively, and NCOA4, FTL, FTH1, and sequestosome-1 (SQSTM1) were increased. In CCs, autolysosomes are a type of secondary lysosome in which the primary lysosome is fused to the outer membrane of an autophagosome.

Although not included in the term of the autolysosomes of the CC, VIM is a DEP associated with autolysosomes. VIM is a type of intermediate diameter filament that anchors and supports intracellular organelles in mesenchymal cells such as fibroblasts and Schwann cells [82]. VIM plays a physiological role in the positioning of autophagosomes and lysosomes and has been shown to be an important factor in the regulation of autophagy. It has been reported that the inhibition of VIM results in the accumulation of autophagosomes and the inhibition of autophagy [83,84]. In the present measurement, VIM and nestin, an intermediate diameter filament that interacts with VIM in neurons, were increased in both the KO1 and KO2 cells. In the NPC1 knockout cells, the phosphorylation of the intermediate filament VIM was decreased compared to the WT cells, and this hypophosphorylation results from reduced protein kinase C activity [85]. It has been reported that the intracellular translocation of LDL-derived cholesterol from lysosomes is a VIM-dependent process, and the activation of protein kinase C solubilizes VIM and eliminates the accumulation of cholesterol [86,87].

GBA and MAN2B1 are both lysosomal hydrogenases. Unlike Niemann–Pick disease types A and B, which are caused by a deficiency in acid sphingomyelinase, deficits in lysosomal enzymes are not direct causes of NPC. However, secondary alterations of lysosomal enzymes in NPC have been reported [88,89], which were identified as DEPs. GBA is responsible for the degradation of glucosylceramide in lysosomes [90]. A marked decrease in activity due to mutations in the *GBA* leads to Gaucher’s disease, a type of LSD like NPC which results in the accumulation of glucosylceramide and secondary cholesterol accumulation [91]. Several reports have shown that the mass of GBA is reduced in NPC [16,92,93], which is consistent with the results of the proteome variation analyses of the KO1 and KO2 cells in this study. It has been reported that in NPC, cholesterol accumulation decreases GBA, and cholesterol depletion restores GBA levels [92]. GBA2, a glucosylceramide hydrolyzing enzyme present outside the lysosome, has been reported to be particularly abundant in Purkinje cells (PCs), one of the neuronal populations most affected by NPC, in a compensatory manner due to reduced GBA. In Npc1 −/− mice, GBA2 was found to be reduced in brain-permeable low-nanomolar inhibitors with significantly improved motor coordination and prolonged lifespan despite no improvement in cholesterol or ganglioside abnormalities. It is suggested that GBA2 activity is a therapeutic target for NPC [94]. Note that although GBA2 was identified in all cell types in this study, no significant changes were observed. In addition, miglustat, an inhibitor of ceramide glucosyltransferase, the enzyme responsible for the synthesis of glucosylceramides, is the only currently approved drug for the treatment of NPC. These results suggest that GBA and GBA2 are involved in pathogenesis in relation to glucosylceramide accumulation in NPC and could even be therapeutic targets.

Lysosomal alpha-mannosidase (MAN2B1) is a lysosomal protein that hydrolyzes the alpha-linked terminal mannose of glycoproteins. Defects in MAN2B1 cause alpha-mannosidosis, which is an autosomal recessive genetic disorder and an LSD like NPC. In alpha-mannosidosis, oligosaccharides including mannose accumulate in lysosomes [95]. Symptoms include mental and cognitive impairments, hearing loss, and ataxia, with a wide range of onset dates and similar symptoms to those observed in NPC [96,97]. MAN2B1 is also known to be associated with Parkinson’s disease, which is a neurodegenerative disease like NPC and has been reported to be a promising biomarker candidate in cerebrospinal fluid from patients with Parkinson’s disease [98]. In another report, MAN2B1 was decreased in the liver of Niemann–Pick-disease-model mice, which is consistent with the results in the KO1 and KO2 cells observed in the present study [99]. Thus, MAN2B1 has been associated with LSDs and neurodegenerative diseases, and the observed decrease in MAN2B1 could have important implications in Niemann–Pick disease.

CD63, also known as lysosomal membrane-associated protein 3 (LAMP3), is found primarily in the inner membrane of late endosomes [100]. It is also abundant in extracellular vesicles, and CD63 regulates the efflux of ferritin–Fe^2+^ complexes bound to the cargo protein NCOA4 via extracellular vesicles [101]. It has been reported that CD63 expression increases in response to intracellular iron loading and decreases in response to deficiency. The expression of CD63 is regulated by iron via the IRE-IRP system, which is responsible for regulating the expression of iron metabolism-associated proteins such as ferritin. Specifically, a canonical IRE in the 5′ untranslated region of CD63 messenger RNA is responsible for regulating its expression in response to increased iron [102,103,104]. In this proteome variation analysis, CD63, FTL, and FTH1 were all predominantly increased, suggesting iron loading in the NPC model cells. This is consistent with previous reports of iron accumulation in the brains of NPC1 knockout mice [71].

Other proteins closely related to NPC were also identified. NPC1 was identified in all replicates in the WT cells but not in all replicates in the two types of KO cells. This is consistent with the intent of creating the cells in this study and is one basis on which the proteome reflects the pathogenesis of NPC. NPC2, one of the proteins responsible for NPC, was increased in both the KO1 and KO2 cells (by 2.1- and 1.8-fold, respectively). This is consistent with the previous proteomic analysis report [105]. Lysosome-associated membrane glycoprotein 1 (LAMP1) and lysosome-associated membrane glycoprotein 2 (LAMP2) showed significant increases (by 1.4- and 1.8-fold for KO1 and 1.4- and 1.3-fold for KO2, respectively), but were not DEPs. LAMP1 and LAMP2 account for approximately 50% of lysosomal membrane proteins and contribute to autophagy and cholesterol homeostasis [106]. LAMP1 and LAMP2 share many functions and play important roles in lysosomal cholesterol homeostasis, especially in the absence of NPC1, by binding to cholesterol and facilitating cholesterol efflux out of the lysosome [107]. These lysosomal proteins are also known in relation to other lysosomal storage diseases, such as Fabry disease, and could be associated with the neurodegenerative symptoms common to these diseases [108,109].

Mitophagy is a mitochondrial-selective form of autophagy for the elimination of damaged mitochondria [110]. The relationship between mitophagy and NPC has been widely reported. In NPC1-deficient cells, the activation of mTORC1 signaling and the associated autophagy are suppressed [111,112,113]. In other reports, the accumulation of cholesterol in lysosomes is observed in NPC which leads to lysosomal enlargement and the dysfunction of autophagosomes and mitochondria [114,115,116]. Abnormal mitochondrial clearance and a lack of mitophagy have been shown in NPC knockdown cells [117]. It was also reported that in NPC1 knockout cells, the inhibition of mTORC1 signaling resolved proteolytic defects in lysosomes and a lack of mitophagy without restoring cholesterol accumulation [118]. In the enrichment analysis, mitophagy was identified in the KEGG pathway in the KO1 and KO2 cells, with increased calcium-binding and coiled-coil domain-containing protein 2 (CALCOCO2), gamma-aminobutyric acid receptor-associated protein-like 2 (GABARAPL2), and MAP1LC3B and decreased SQSTM1. In BPs, macromitophagy was also identified at the eighth and twenty-eighth terms in the KO1 and KO2 cells, respectively, with increased MAP1LC3B and decreased SQSTM1 and GBA. Macromitophagy is defined as a selective autophagy process in which mitochondria are degraded by macroautophagy in a BP. CALCOCO2 is an autophagy adaptor, specifically known as a loading receptor for xenophagy [119]. On the other hand, it has been reported to mediate autophagosome maturation by binding to LC3B, GABARAPL2, and Myosin-VI (MYO6) in non-infected cells, suggesting that CALCOCO2 is not only involved in targeting bacterial autophagosomes, including all non-xenophagy autophagy [120]. GABARAPL2 is a member of the Atg8 orthologue as well as MAP1LC3B and is also used in mitophagy [121,122]. It has also been reported to be involved in autophagosome maturation in mitophagy in concert with CALCOCO2 and others, as mentioned above. SQSTM1 is an autophagy receptor like CALCOCO2 and others which is involved in mitophagy by recognizing ubiquitinated mitochondria together with CALCOCO2 [123,124]. Mitochondrial dysfunction based on the loss of mitophagy has been reported in knockin mice with allelic mutations of GBA [125]. Decreased GBA could be one of the causes of the impaired degradation of NPC in lysosomes and, as mentioned above, the inhibition of mTORC1 could restore function and mitophagy. Although the relationship between the increase or decrease in DEPs and the widely reported lack of mitophagy in NPC is not clear, the fact that a mitophagy-related term was extracted at a high rank on the list in this study supports the association between NPC and mitophagy. In addition, we also reported a decrease in steroid hormone metabolism related to mitochondrial dysfunction [126] in which the steroid biosynthesis pathway was significantly altered in the KO1 cells (Figure 5a and Appendix A).

This study was analyzed using NPC-model hepatocellular carcinoma cells, and the liver plays a pivotal role in metabolic pathways. Therefore, it is possible that there are proteins in the metabolic pathway that are closely associated with hepatosplenomegaly and other liver lesions. On the other hand, this enrichment analysis did not show any term associated with liver damage in either the GO or KEGG. In addition, many studies have been published regarding metabolic alterations, and we focused on lipid metabolism and reported its use as a diagnostic biomarker [4,5,6,7]. Amino acid alterations were reported by other groups [127,128].

There are a few proteome analyses using NPC-model cells or animals. In the two reports that analyzed the proteome using hepatocytes from NPC1 knockout mice, annexin A1, catechol-O methyltransferase (COMT), and lysosomal proteins such as cathepsin B (CTSB), cathepsin D (CTSD), and ubiquitin-like-conjugating enzyme ATG3 (ATG3) were identified as DEPs, but they were not DEPs in this analysis, although each protein was identified [129,130]. On the other hand, several common terms were identified for the GO in two reports and this analysis, including autophagy, iron homeostasis, and the heme synthesis process. In neuroblastoma cells reacted with U18666A, a process which causes symptoms similar to NPC, a number of common GO terms were identified from the top to the bottom, including the sterol biosynthesis process, cholesterol biosynthesis process, and lipid metabolic process [131]. In a proteome analysis of the cerebella of NPC1 knockout mice, lysosomal proteins such as beta-hexosaminidase subunit alpha (HEXA), beta-hexosaminidase subunit beta (HEXB), lysosomal acid lipase/cholesteryl ester hydrolase (LIPA), MAN2B1, sialidase-1 (NEU1), prosaposin (PSAP), and tripeptidyl-peptidase 1 (TPP1) were all identified as DEPs, but in the present study, none of them were identified as DEPs which showed only a slight increase or decrease except for MAN2B1. Since these measurements were performed via the LF method and not via quantification, the reproducibility between measurements for each protein is limited. Although the directions of the increase/decrease were in good agreement, the proteins detected as DEPs in these assays only suggest that they could be involved in NPC. On the other hand, a pathway analysis showed that many of the pathways were highly consistent regardless of the region being analyzed, such as the liver or brain. In the pathway analysis, rankings are determined based on variations in the results of multiple proteins, so it is possible that the LF analysis also showed results with a certain degree of reliability. In the future, therapeutic targeting should be performed based on these results.

### 2.4. The Accumulation of Lipid Peroxide Was Observed in Two Types of NPC Model Cells

The LF proteome variation analysis and subsequent enrichment analysis suggested a relationship between ferroptosis and NPC model cells. However, it is not clear whether ferroptosis progresses in the NPC model cells. Therefore, we measured the fluorescence induced by lipid peroxide, which characteristically accumulates with the progression of ferroptosis [77,132]. We used *tert*-butyl hydroperoxide (TBHP, 500 μM) which induces lipid peroxidation and ferroptosis, as a positive control [133], and Liperfluo, which is widely used in the study of ferroptosis due to its lipid peroxide-specific fluorescence emission [134,135] (Appendix A). The results showed that the fluorescence intensities of both the KO1 and KO2 cells were significantly increased compared to the WT cells. Lipid peroxide accumulation was observed in the two types of NPC model cells, suggesting that ferroptosis could be underway in NPC.

On the basis of iron accumulation in NPC knockout mice, meanwhile, it has been reported that treatment with deferiprone, an iron chelator, did not extend lifespan or restore symptoms [136]. This was contrary to the results of iron chelation therapy in AD and PD, neurodegenerative diseases in which the same iron accumulation was observed, leading to neurological improvement [137,138]. However, NPC differs from these diseases in that it often develops in childhood [89]. Iron is essential for brain development, and balanced iron levels are needed, especially during growth [139]. In mice, excessive iron removal at an age of 3 months, equivalent to a human age of about 13.4 years, could lead to iron depletion during growth and a lack of therapeutic efficacy in mice. Thus, regulated iron removal or the inhibition of ferroptosis itself could be effective therapeutic targets in NPC.

There are several limitations to this study. First, further experimental validation is needed to verify the involvement of the DEPs identified in this study with molecular processes and pathways in NPC model cells. In particular, further validation is needed to demonstrate the involvement of ferroptosis, not only by showing an increase in intracellular lipid peroxides but also by showing intracellular Fe^2+^, the accumulation of ROS, and so on [134,140]. Second, since we used an in vitro model of NPC, future studies are needed to verify the roles of DEPs and the pathways analyzed from them in the pathogenesis of NPC and their potential as therapeutic targets.

## 3. Materials and Methods

### 3.1. Chemicals and Reagents

Ultrapure water was prepared using a Puric-α apparatus (Organo Corporation, Tokyo, Japan). Acetonitrile was purchased from Kanto Kagaku (Tokyo, Japan); formic acid, methanol, and chloroform were purchased from FUJIFILM Wako Pure Chemical Co. Ltd. (Osaka, Japan); and the nano-HPLC ODS capillary column (75 μm × 12.5 cm, 3 μm) was acquired from Nikkyo Technos (Tokyo, Japan). The iodoacetamide (IAA), sodium dodecyl sulfate (SDS), tris(2-carboxyethyl) phosphine (TCEP), triethylammonium bicarbonate (TEAB), trypsin, Dulbecco’s modified Eagle’s medium (DMEM), and non-essential amino acid (NEAA) mixture reagents were obtained from Nacalai Tesque, Inc. (Kyoto, Japan). The Acclaim PepMap 100 trapping column (75 μm × 2 cm, 3 μm), M-PER^™^ Mammalian Protein Extraction Reagent, Pierce^™^ 660 nm Protein Assay Kit, Pierce^™^ Detergent Removal Spin Columns, and Pierce^™^ Peptide Desalting Spin Columns were purchased from Thermo Fisher Scientific (Waltham, MA, USA). TBHP was purchased from Tokyo Chemical Industry Co., Ltd. (Tokyo, Japan), and Liperfluo was purchased from DOJINDO LABORATORIES (Kumamoto, Japan).

### 3.2. LC/MS/MS Equipment, General Conditions, Data Acquisition, and Data Analysis

An EASY-nLC (Thermo Fisher Scientific) was connected to a quadrupole ion trap and an Orbitrap Fusion Tribrid tandem mass spectrometer equipped with an electrospray ionization probe (Thermo Fisher Scientific). The electrospray voltage and ion transfer tube temperature were set to 2000 V and 275 °C, respectively. The MS scan range, resolution, maximum injection time, and RF lens were set at *m*/*z* 375–1600, 60,000 Da, 50 ms, and 60%, respectively. Data acquisition was performed in the data-dependent analysis (DDA) mode under positive ion detection. High-collision dissociation (HCD) was used as the activation type. For the MS/MS analysis, the dynamic exclusion duration, intensity threshold, isolation window, HCD collision energy, and maximum injection time were set at 20 ms, 5 × 103 cps, 1.6 Da, 30%, and 35 ms, respectively.

Mixtures of formic acid/water (0.1:100, *v*/*v*) and formic acid/water/acetonitrile (0.1:20:80, *v*/*v*/*v*) were used as mobile phases A and B, respectively. The flow rate was set at 300 nL/min. A Nano HPLC capillary ODS column (75 μm i.d. × 12.5 cm, 3 μm; Nikkyo Technos) and Acclaim PepMap 100 (75 μm i.d. × 2 cm, 3 μm; Thermo Fisher Scientific) were used as the analytical and trapping columns, respectively. The equilibration of the analytical and trapping columns was performed using flows of 5 and 7 μL of the initial mobile phase, respectively.

Data acquisition was performed using Xcalibur (version 4.3, Thermo Fisher Scientific, accessed on 10 June 2022) and Proteome Discoverer (version 2.5.0.400, Thermo Fisher Scientific, accessed on 10 June 2022) for data integration. Proteins were identified using the UniProt database (Version 2022_05, accessed on 10 June 2022). The maximum number of cleavage misses was set to 2. The carbamidomethylation of cysteine was considered a fixed modification, and the oxidation of methionine and the N-terminal acetylation of the protein were considered variable modifications. Proteins were identified with a false discovery rate (FDR) of less than 1%. Measurements for each sample were standardized via summed intensity. When comparing two cell types, proteins with a missing value in more than 20% of replicates from one cell type were excluded from the analysis. A PCA was performed using Metaboanalyst (Version 5.0, set to default, accessed on 11 October 2023). PPI networks were constructed using STRING (version 11.0, with a minimum required interaction score set to medium confidence (0.4), accessed on 10 October 2023) [141]. The GO and KEGG enrichment analyses were carried out using the DAVID database (version 2022_04, set to default, accessed on 14 February 2023) [142].

### 3.3. The Establishment of NPC Cell Models 

NPC cell models were developed using the CRISPR/Cas9method [53,54,55] summarized in Appendix A. Hep G2 cells (WT) were used in this study. The NPC1 gene located on chromosome 18 (18q11.2.) contains 80,715 bases and 25 exons (Appendix A) [57]. In this study, two types of knockout model cells were established by targeting multiple sites of the NPC1 gene (Appendix A). The NPC KO model cell KO1 was developed using two sgRNAs targeting 238–257 bp’s on exon 1 (Site A) and 1357–1376 bp’s on exon 8 (Site C) (Appendix A). The other model (KO2) was developed using two sgRNAs targeting 279–301 bp’s on exon 1 (Site B) and 3589–3611 bp’s on exon 22 (Site D) (Appendix A). Using GGGenome (https://gggenome.dbcls.jp/ja/, accessed on 27 March 2019) and UCSC Genome Browser (https://genome.ucsc.edu/, accessed on 27 March 2019) software, we selected each CRISPR targeting site of the NPC1 gene which showed no homology with other genes. HepG2 WT cells were seeded in 10 cm dishes on day 0 and transfected with px330-based CRISPR/Cas9 vectors (Appendix A) twice on days 2 and 3. On day 5, cell cloning was performed. From the cells targeting Site A and Site C, 22 clones were obtained, and KO1 was selected as a typical clone. From the cells targeting Site B and Site D, 18 clones were obtained, and KO2 was selected as a typical clone. From genomic sequencing, KO1 has a deletion of g at nt1373, an insertion of g after nt1373, and a large deletion between nt241 and nt1373, and KO2 has a tc deletion at nt283 and nt284, an insertion of g after nt283, and a large deletion between nt284 and nt3593. For the confirmation of NPC1 KO, an immunoblot analysis using an NPC1 antibody (#ab55706, Abcam, Cambridge, UK) [118,143], filipin staining [13,56], and cholesterol quantification with cholesterol E-test Wako [126] were performed (Appendix A and Figure 1).

### 3.4. Cell Culture and Cellular Protein Extraction

The WT, KO1, and KO2 cells developed were cultured in DMEM containing 10% FBS, penicillin–streptomycin, and NEAA. The 2.0 × 10^6^ cells were seeded in a 100 mm Petri dish and grown in the medium. After confluent cultures, the cells were washed twice with 3 mL of PBS and subsequently removed from the Petri dish using a scraper in 10 mL of PBS. The suspended cells were counted using a cell counter, and equal numbers of cells were frozen.

The frozen cells were mixed with 1 mL of M-PER^™^ Mammalian Protein Extraction Reagent. The mixture was shaken for 10 min and centrifuged at 14,000× *g* at 4 °C for 15 min. The supernatant was transferred to another tube, and the protein concentrations were quantified using the Pierce™ 660 nm Protein Assay Kit.

A sample of the proteins (100 μg) was transferred into a separate tube and adjusted to 100 μL with a mixture of 0.1% SDS and a 100 mM TEAB aqueous solution. In succession, 5.3 μL of a mixture of 20 mM TCEP in 0.1% SDS and 100 mM TEAB was added, and the mixed solution was heated at 95 °C for 10 min. Subsequently, 1.7 μL of a mixture of 500 mM IAA in 0.1% SDS and 100 mM TEAB was added, and the solution was maintained at 25 °C for 60 min in the dark.

To this alkylated protein fraction, 428 μL of methanol, 107 μL of chloroform, and 321 μL of water were added, and the solution was mixed and centrifuged at 15,000× *g* at 25 °C for 1 min. The supernatant was removed, and 428 μL of methanol was added to the remaining solution. The mixture was again centrifuged at 15,000× *g* at 25 °C for 1 min, and the supernatant was removed. The remaining material was dried at 25 °C using a CC-105 centrifugal concentrator (TOMY, Tokyo, Japan). The dried pellet was dissolved in 100 μL of 0.1% SDS and 100 mM TEAB. To the solution, 4 μL of 0.5 μg/μL trypsin aqueous solution was added, and the mixture was incubated at 300 rpm at 37 °C overnight. The mixture was then sequentially placed onto a Pierce^™^ Detergent Removal Spin Column and Pierce^™^ Peptide Desalting Spin Column, and the eluted solution was dried using a CC-105 centrifugal concentrator. The pellet was dissolved in 200 μL of a mixture of 0.1% formic acid and 2% acetonitrile in water, and the solution was used as the sample (protein amount: approximately 0.5 μg/μL).

### 3.5. nLC Condition for Label-Free Global Proteomics

The gradient program was as follows: 0–5% B over 1 min, 5–40% B over 60 min, 40–95% B over 2 min, and 95% B over 17 min (cycle time: 80 min). The injection protein volume and amount were 2 μL and 1 μg, respectively. The injected sample solution was diluted with mobile phase A up to 5 μL and applied to the trapping column. Five cell dishes were prepared for each cell line and analyzed.

### 3.6. Cell Fluorescence Intensity Measurement

WT, KO1, and KO2 cells (40,000 cells per well) were seeded in 96-well plates, with only DMEM medium (with serum) as a background, the WT cells as a control and positive control, and the KO1 and KO2 cells as comparisons, respectively. After overnight incubation in a tissue culture incubator at 37 °C, the cells were washed with serum-free DMEM medium, 70 µL of 10 µM Liperfluo was added, and the cells incubated for 30 min. After washing the cells twice with Hanks’ Balanced Salt Solution (HBSS), HBSS was added for background and samples, and 500 µM TBHP was added for the positive control and incubated for 1 h. After washing the samples with HBSS twice, they were measured using an Infinite^®^ M200 PRO (Tecan, Männedorf, Switzerland) at an excitation wavelength of 488 nm and an emission wavelength of 535 nm. The mode was set to bottom, the gain was set to optimal, and the number of flashes was set to 10. An analysis was performed by eliminating background data due to the culture medium and excluding samples with bubbles or samples with fluorescence intensities that were more negative than the background. Standard deviations were applied to error bars, and *p* values were derived using the two-tailed, unpaired Welch’s *t*-test.

## 4. Conclusions

Here, we aimed to elucidate the pathophysiological aspects of proteomic expression. We developed NPC1 model cells using the CRISPR/Cas9 method, and global proteomic approaches were used. In result, many differentially expressed proteins were identified using the label-free method, and the two cell models exhibited different alterations; however, commonly changing proteins were also observed. To identify the differential pathways in the pathology of NPC, we performed a bioinformatics analysis. The identification of NCOA4, FTH1, FTL, and various autophagy-related proteins as DEPs and the enrichment of ferroptosis in the pathway analysis could suggest a relation between NPC and ferroptosis which has rarely been reported. Associations between neurodegenerative diseases and ferroptosis have been reported in many diseases, and our results also suggest an association with NPC, which presents with clinical neurodegenerative symptoms. In addition, other LSD-causing genes, such as the lysosomal enzymes GBA and MAN2B1 and the lysosomal membrane protein CD63, were identified as DEPs, consistent with previous reports and supporting the association with other LSDs and reports of abnormal iron metabolism. These pathways and proteins could serve as novel therapeutic targets, and therapeutic research based on these results is expected in the future.

## Figures and Tables

**Figure 1 ijms-24-15642-f001:**
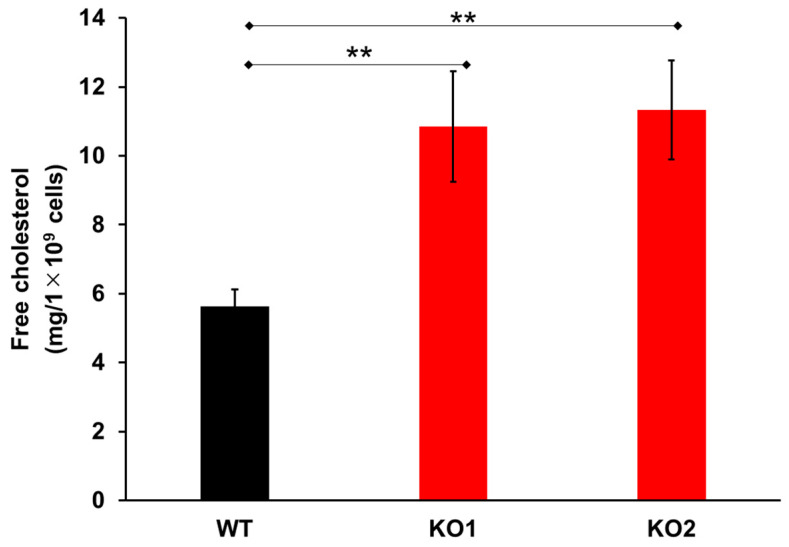
Accumulated cholesterol quantitation in wild-type cells and two types of NPC1 knockout cells. Data represent the means ± SDs, *n* = 9, ** *p* < 0.01 (two-tailed Welch’s *t*-test). WT—Hep G2 cells; KO1—Sites A and C mutant NPC1 model cells; KO2—Sites B and D mutant NPC1 model cells.

**Figure 2 ijms-24-15642-f002:**
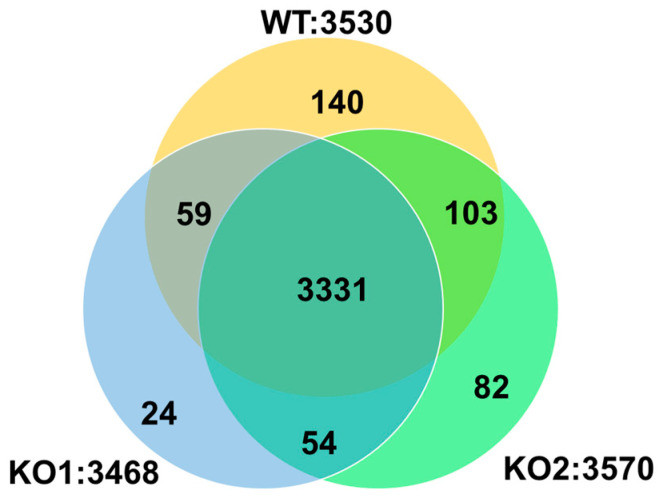
Venn diagram showing protein numbers identified from the results of the label-free global proteomics analysis. WT—Hep G2 cells; KO1—Sites A and C mutant NPC1 model cells; KO2—Sites B and D mutant NPC1 model cells.

**Figure 3 ijms-24-15642-f003:**
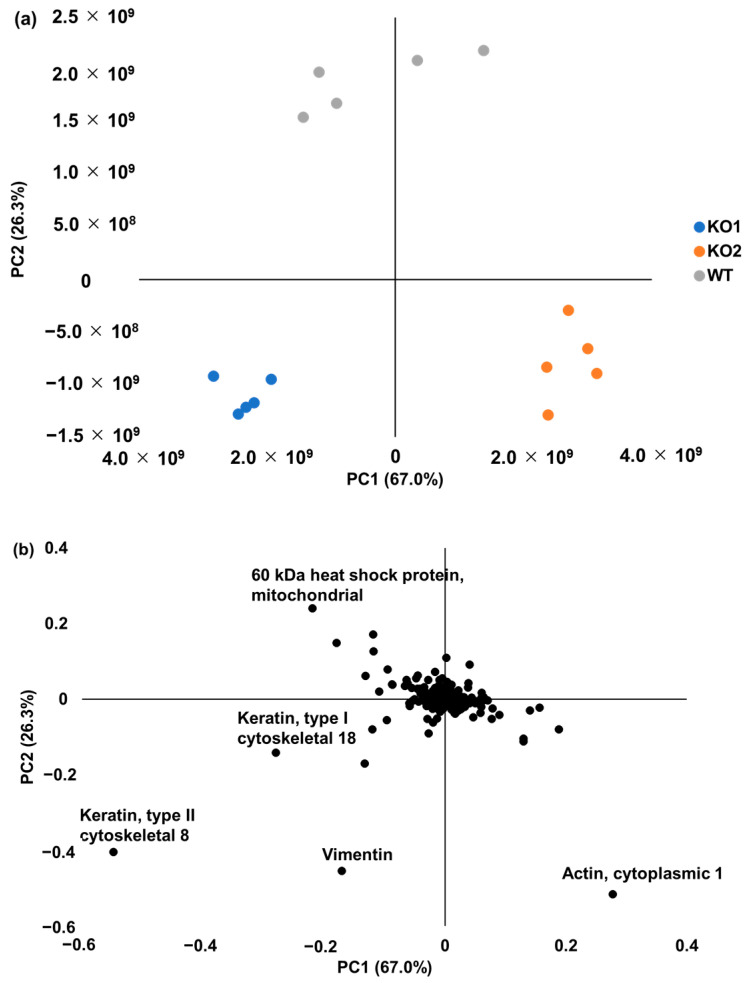
The principal component analysis of the LF method. (**a**) A PCA plot using 3331 proteins identified in the KO1, KO2, and WT cells. (**b**) A loading plot using 3331 proteins identified in the KO1, KO2, and WT cells. WT—Hep G2 cells; KO1—Sites A and C mutant NPC1 model cells; KO2—Sites B and D mutant NPC1 model cells.

**Figure 4 ijms-24-15642-f004:**
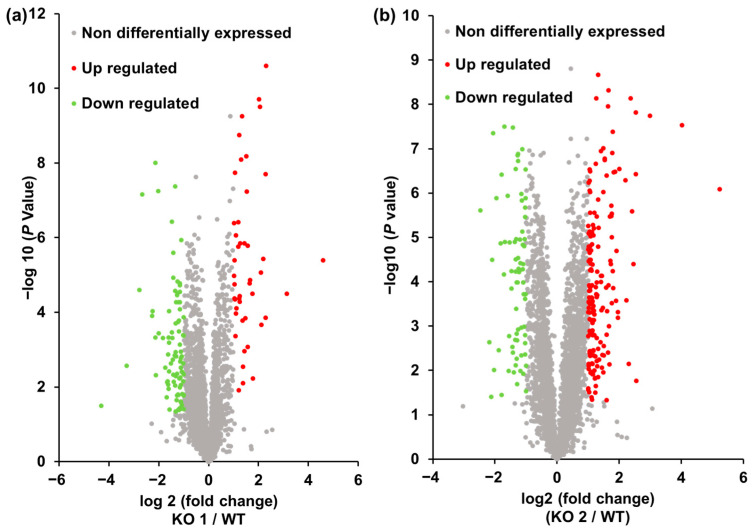
Volcano plots from the results of the label-free global proteomics analysis. (**a**) KO1 versus WT using 3390 proteins identified in KO1 and WT cells. (**b**) KO2 versus WT using 3434 proteins identified in KO2 and WT cells. Red-colored plots—greater than 2-fold upregulated and *p* < 0.05 (adjusted using the Benjamini–Hochberg correction) proteins in NPC1 model cells; green-colored plots—greater than 0.5-fold downregulated and *p* < 0.05 (adjusted using the Benjamini–Hochberg correction) proteins in NPC1 model cells; grey-colored plots—non-differentially changed proteins. WT—Hep G2 cells; KO1—Sites A and C mutant NPC1 model cells; KO2—Sites B and D mutant NPC1 model cells.

**Figure 5 ijms-24-15642-f005:**
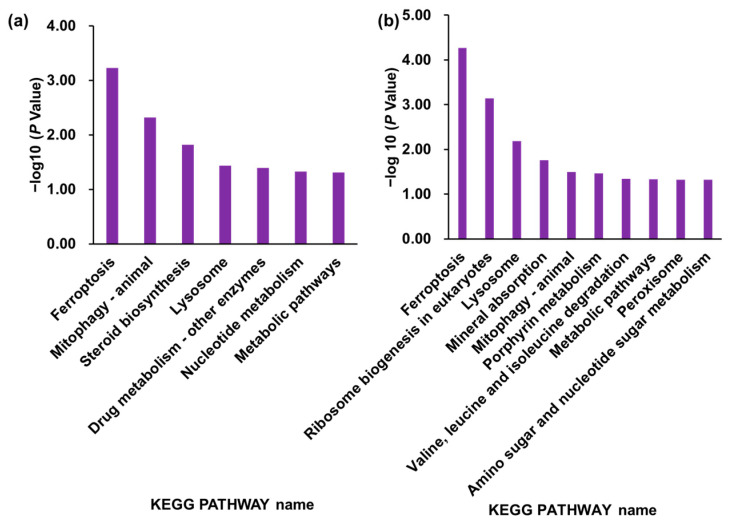
Enrichment analysis in the KEGG pathway from the results of the label-free global proteomics analysis. (**a**) KO1 versus WT using 136 DEPs in KO1. (**b**) KO2 versus WT using 249 DEPs in KO2. Proteins with *p* < 0.05 (Fisher’s exact test) are shown in these figures. WT—Hep G2 cells; KO1—Sites A and C mutant NPC1 model cells; KO2—Sites B and D mutant NPC1 model cells.

**Table 1 ijms-24-15642-t001:** Summary of protein numbers of DEPs in the label-free global proteomics analysis.

	Upregulated DEPs (Numbers)		Downregulated DEPs (Numbers)		Total (Numbers)
Identified in KO1 (numbers)	18		77		95
	25	(a) 2	(b) 4	10	35 (41)
Identified in KO2 (numbers)		(b) 4	(a) 2		
	145		63		208
Total	188	6		150	338 (344)

(a) are identified as upregulated DEPs in KO1 cells and downregulated DEPs in KO2 cells, (b) are identified as downregulated DEPs in KO1 cells and upregulated DEPs in KO2 cells. WT—hep G2 cells; KO1—Sites A and C mutant NPC1 model cells; KO1—Sites B and D mutant NPC1 model cells.

## Data Availability

Data is contained within the article or Appendix A.

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
