# Peer review of "Global Proteomics for Identifying the Alteration Pathway of Niemann–Pick Disease Type C Using Hepatic Cell Models"

_ijms, 2023, doi:10.3390/ijms242115642_

Round 1

Reviewer 1 Report (New Reviewer)

Comments and Suggestions for Authors

Summary:
The manuscript explores the pathophysiology of Niemann–Pick disease type C (NPC) using a global proteomics approach in NPC model cells. It discusses the successful gene editing of NPC1 in these cells and the subsequent proteome variations observed. The study identifies potential therapeutic targets and pathways related to NPC's ferroptosis, lysosomes, and mitophagy. The study also suggests a link between NPC and these pathways. It highlights the diverse proteomic expression patterns in different NPC mutations.

English Quality:
The English in the manuscript is generally clear and well-structured, making it easy to follow the research findings and methodology. However, the manuscript should undergo thorough proofreading to correct minor typographical errors. Additionally, certain complex sentences could be simplified, and some sections might be made more concise while retaining essential information.

Publishability:
Overall, the study is a valuable contribution to understanding NPC at the proteomic level, and the authors have effectively presented their findings. After the changes I propose (i.e., improving statistical/bioinformatics and better contextualizing the findings in current literature), this manuscript has the potential for publication.

Section-specific comments:
Results and discussion: the section opens without an explicit definition or explanation provided by the authors for the acronyms "KO1" and "KO2". I kindly request that the authors provide a clear and concise definition or explanation for the acronyms 'KO1' and 'KO2' when first introduced. This addition will help improve the overall clarity and comprehension of the research paper.
The rationale behind target site selection for CRISPR/Cas9 knockouts is unclear. Besides, the manuscript needs more clarity on sample sizes and whether single clones were used. The authors must address these critical issues as they raise concerns about experimental design and reproducibility.
The manuscript requires more robust evidence to establish a clear link between NPC and the observed proteomic differences. Investigating whether miglustat treatment improves phenotypic aspects like cholesterol accumulation and assessing its impact on proteomic profiles would be beneficial.
The observation that both knockout (KO) cell models share alterations in lysosomal pathways is intriguing, as dysregulation of these pathways has been recognized as a hallmark of lysosomal storage disorders. Notably, diseases like Fabry have exhibited differential expression of lysosomal genes, including enzymes and proteins such as LAMP-1 and LAMP-2. It is crucial to enhance the discussion by citing and comparing recent relevant papers in the field (e.g., doi.org/10.3390/ijms222111339, doi.org/10.3390/ijms24021095, 10.1016/j.gene.2013.11.063) with the findings obtained in this study.

Conclusion: It would be beneficial if the conclusion provided more detail on the specific proteins or genes involved in these pathways and their relevance to NPC pathology, as this would enhance the clarity of the paper's key takeaways.

Line-specific comments:
160-172 I request that you report the percentage or number of PPIs in your network derived from text-mining results within the STRING database. Text-mining-based interactions are generally of lower confidence compared to experimental or curated interactions. Including this information will provide readers and reviewers with a more comprehensive understanding of the reliability of the PPI network you constructed.

Reproducibility checklist:
1. Please report IDs for all biological entities (i.e., genes, proteins, variants) wherever they exist
2. Please report the access date for all web-based tools and databases.
3. Please report the version number for all software tools.
4. Please report all the non-default choices made when using bioinformatics tools. If only defaults were used, please write that explicitly.

Remarks on statistics:
* Sample size is reported only for some experiments/plots. The authors should integrate this information for all experiments/plots and remember that anecdotal evidence is not admissible.
* I request that the authors address the issue of alpha inflation in their statistical analysis. Since they are comparing multiple group means using plain t-tests, there is an increased risk of false positives. I recommend implementing specific tests like the Tukey correction or similar methods for multiple comparisons to mitigate this risk. This adjustment will enhance the reliability of the statistical analysis.
* The manuscript should clarify whether the p-values presented and used as the basis for their conclusions have been adjusted for multiple comparisons. If they have been adjusted, the specific method and adjusted p-values should be reported. Alternatively, the use of non-adjusted p-values should be explicitly mentioned.

Author Response

Reviewer 2 Report (New Reviewer)

Comments and Suggestions for Authors

The present article entitled: “Global proteomics for identifying the alteration pathway using 2 model cells of Niemann–Pick disease type C” by Keitaro Miyoshi et al. is interesting with novel results. It is however unfortunate that the authors didn’t use the existing proteomic data publicly available to compare their results. Such data include samples from Npc1 mice or other in vitro models based on cancer cell lines. Comparing the results with the existing one, notably in vivo, would strengthen the observation. 

I have few suggestions to potential improve the interest of the present manuscript:

Previous reports have discussed hepatocellular carcinoma in NPC (PMID: 36636588). The experimental design suggests the study of NPC1 gene in the context of liver cancer. This should be further highlighted in the intro and discussion of the manuscript. Could the finding of this manuscript be a driver for hepatocellular carcinoma?

Could the difference see between clone be driven by off target of the Crispr KO?

Liver proteomic studies have been performed in Npc1 mice (PMID: 30870990; PMID: 34440927). How do your results compare to the mouse results? Oxidative stress was previously highlighted in Npc1 mice liver cells. As this is an integral part of the ferroptosis pathway these previous results should be discussed.

The inhibition of ferroptosis using deferiprone led to reduced lifespan in Npc1 mice. These results should be discussed (PMID: 32100150).  

What is displayed on the graph in figure 1 should be understandable with the legend. Mean and standard deviation? number of samples?

The proteomic is not at the single cell resolution but rather performed on many cells, this should be clear from the text.

Figure 2a is a Venn diagram.

The PCA results are a major part of the data analysis and should be part of the main figures not in supplemental data. What are the proteins driving the PC1 and 2 separations?

Volcano plot is just a way to display differential expression analysis results not an analysis. A potential visual way to show the difference between KO1 and KO2 normalized to WT could be to plot on the x-axis log2FC KO1vsWT and y-axis log2FC KO2vsWT. The protein specifical enriched or decreased in one KO will appear as outliers.   

Proteomic was also performed on neuroblastoma cells used as model of NPC (PMID: 36920149). How do the author results compare to this work also using cancer cells?

Line 198: “lipid peroxidation of lipids” of lipids can be deleted.

Ferroptosis involvement in NPC was previously suggested in PMID: 33419148. This reference should be included.

Line 328: the references are spited into two brackets.

 The premise of the article is about liver disease however NPC is also a neurodegenerative disease. How do the proteomic on liver cancer cells compare to mouse brain? PMID: 23070805. Similarities or differences could be included to highlight the tissue specific proteomic change.

The title is misleading, it should be clearly stated that the manuscript is about NPC in the context of liver diseases.

Round 2

Reviewer 1 Report (New Reviewer)

Comments and Suggestions for Authors

Paper is now suitable for publication

Reviewer 2 Report (New Reviewer)

Comments and Suggestions for Authors

I would like to thank the authors for addressing my comments and answering my questions. I believe the inclusion of the comparison with previous dataset improves the quality of the manuscript. It is interesting that many of the pathways identified are conserved across models independently of the technology used for proteomic analysis. These similarities further strengthen the choice of the author to build this model.

This manuscript is a resubmission of an earlier submission. The following is a list of the peer review reports and author responses from that submission.

Round 1

Reviewer 1 Report

Comments and Suggestions for Authors

The manuscript by Miyoshi et al. presents proteomic studies performed on cellular models of Niemann-Pick disease type C (NPC). CRISPR/Cas9 method was used to introduce mutations in one of the genes associated with NPC, NPC1, thus generating cells lacking its expression and showing cholesterol accumulation. Having generated a cellular model of the disease, the authors performed proteomic analyses using two different methods in order to identify proteins whose expression is altered in NPC1-mutated cells compared to wild type in the effort to identify novel molecules that could represent therapeutic targets.

Major comments

The paper is generally well written and results are presented in a sufficiently clear way. However, the overall study fails to deliver strong conclusions about its findings providing only an analysis of altered pathways emerging from the proteomic analyses. My feeling is that the study would considerably gain significance if some of the identified differentially expressed proteins were experimentally validated and if alteration of the pathways as inferred by proteomic analyses could be proved experimentally, also availing of the generated cellular models which recapitulate the disease, in order to lay stronger basis to the understanding of the pathophysiology of NPC. Furthermore, how do the author comment about the limited overlap between results obtained with the two different MS methods?

Minor comments

·    It is not clear to me why the authors decided to target two different regions of the NPC1 gene for generation of each knock out cell line. Did the authors check that in the selected clones mutations were introduced in correspondence of the binding regions of both gRNA used? Have the authors checked for the expression of truncated forms of NPC1 in case mutations were introduced only upon targeting by the site C and site D gRNAs, since these target regions in exon 8 and exon 22, respectively?

·    Of the 22 and 18 clones obtained for KO1 and KO2, respectively, was only one for each genotype selected for experiments, or did the authors use a pool of KO1 and KO2 clones?

·    In figure 2 and figure 4, the authors possibly meant to write “Venn” diagram? Furthermore, the legends do not clearly indicate what the letters a-f stands for (this is vaguely inferable by looking at the numbers in tables S1 and S3)

·    In figure 3 legend the authors referred to red or green plots but I think they meant to write dots. Besides, the proteins represented by grey dots should be referred to as “non differentially expressed” in the graph legend and not just as “proteins”

·    Table 1 (and in lower instance table 2) are not properly explained and I had difficulties in understanding the meaning of the values represented. For instance lines 137-141 describes table 1: “Twenty-four proteins were upregulated in both KO1 and KO2 and 10 proteins were downregulated” – are these identical proteins identified in the two KOs? – “Two proteins increased in KO1 but decreased in KO2, and four proteins decreased in both KO1 and KO2.” – I think here the authors refer to proteins represented with the symbols a, a’, b and b’ (although these symbols are never explained in the manuscript nor the table legend)? Furthermore, I thought the 4 proteins mentioned were upregulated in KO2 and downregulated in KO1, while the number of proteins downregulated in both KOs is 10?

·    In figure 4 legend there is also the legend for figure 5 (again with the mentioning of plots instead of dots)

·    In legends for figures S4, S5, S6 and S7 KO1 is indicated instead of KO2

Reviewer 2 Report

Comments and Suggestions for Authors

The authors presented a MS-based proteomics investigation on the differentially expressed proteins in NPC1/NPC2-caused NPC disease using CRISPR-edited cell model. This work provided a proteomics landscape of the NPC. However, a series of flaws should be improved:

1. The authors created NPC1/NPC2-defected cell models. It seems that these models are created in this study. There are two major problems:

(1) Are these models reflecting real diseases? Since the real NPC1/2 defects contains many possibilities (various mutations), the authors should justify that the created cell models reflect real cases.

(2) The authors should describe the details of the CRISPR/Cas9 editing and the evidence of successful editing, not just presenting a diagram showing that the cholesterol up-regulation.

2. I'm wondering why the authors performed two types of MS proteomics: lable-free and TMT. The results of the two methods difffer largely. The authors should justify the technical quality of these two experiments.

3. There is no quality control of the MS experiments and data processing. Please refer to the Human Proteome Project Guideline.

4. After the MS, the authors simply did a KEGG pathway enrichment of the differentially expressed proteins of KO1 and KO2. No further analysis was performed. So what do the pathways enrichment mean? What is the novel insights to the disease?

5. The English is very hard to follow. The authors should ask a native speaker to proofread the entire manuscript.